

# Collisional mechanics of the diagonal gaits of horses over a range of speeds

Sarah Jane Hobbs[1] and Hilary M. Clayton[2]

[1] University of Central Lancashire, Preston, United Kingdom
[2] Sport Horse Science, Mason, MI, United States of America

## ABSTRACT

One of the goals of the neuromotor control system is to minimize the cost of locomotion by reducing mechanical energy losses. Collisional mechanics, which studies the redirection of the downwards motion of the center of mass (COM) by ground reaction forces (GRF) generated by the limbs, represents an important source of energy loss. The primary objective of this study was to compare collisional mechanics and the associated mechanical energy losses in horses performing diagonally-synchronized gaits over a range of speeds. It is to be expected that collisional energy losses will be high when the COM velocity vector is closely aligned with the GRF vector. This condition is achieved in piaffe, an artificial gait performed in dressage competitions that has a diagonal limb coordination pattern similar to trot but performed with little or no forward velocity. Therefore, we hypothesized that collisional energy losses would be higher in piaffe than in trot. Synchronized kinematic and GRF data were collected from three highly-trained horses performing piaffe, passage and trot at a range of speeds. Derived variables were vertical excursion and velocity of the trunk COM, fore and hind limb compression expressed as percentage reduction of standing limb lengths, range of limb pro-retraction, GRF vector magnitude and vector angle, collision angle ($\Phi$), and mechanical cost of motion ($CoMot_{mech}$). Linear regression was used to investigate the relationship between $CoMot_{mech}$ and speed for each gait. Partial correlation was used to seek relationships between COM excursion and limb mechanics for each gait. Piaffe, passage and trot were clearly separated on the basis of speed. In all gaits the trunk was high at contact and lift off and descended to its lowest point in midstance following the pattern typical of spring mass mechanics. Mechanical cost was significantly ($p < .05$) and inversely related to speed in trot and piaffe with the value increasing steeply as speed approached zero due to a near vertical orientation of both the COM velocity vector and the GRF vector. Limb compression during stance was significantly ($p < .05$) linked to trunk COM vertical excursion in all gaits, with a stronger relationship in the forelimb. Hindlimb compression was, however, large in piaffe where the force magnitudes are notably smaller. The study illustrates the potential value of studying artificial gaits to provide data encompassing the entire range of locomotor capabilities. The results supported the experimental hypothesis by showing a threefold increase in collisional energy losses in piaffe compared with trot. In all gaits, dissociation between diagonal limb contacts and lift offs was thought to be an important strategy in reducing in collisional losses. Piaffe, the most costly gait, has similar characteristics to hopping on the spot. It appears that greater hindlimb compliance and a lower step frequency are important energy conservation strategies for piaffe.

Corresponding author
Sarah Jane Hobbs,
sjhobbs1@uclan.ac.uk

## INTRODUCTION

Locomotion is the act of moving the body, represented by the center of mass (COM), from place to place in relation to the environment. Quadrupeds achieve this by swinging the limbs back and forth, alternating between stance and swing phases. During their stance phases the feet press against the ground to develop ground reaction forces (GRF) that control movements of the COM (*Hobbs & Clayton, 2013*). The mechanical work is broadly divided into internal work used to swing the limbs relative to the body and external work used to move the body, represented by the COM, relative to the environment (*Minetti et al., 1999*). Movements of the limbs and body are driven by muscular contractions that are fueled by the conversion of chemical energy into mechanical energy. There is benefit to minimising energy expenditure for a given movement and conversions between different types of mechanical energy fulfil this function (*Cavagna, Heglund & Taylor, 1977*). The mechanisms of inverted pendulum and spring mass mechanics are based on exchanges between potential energy ($E_p$), kinetic energy ($E_k$) and elastic energy which are all types of mechanical energy. The exchanges occur cyclically during the stride to control the vertical excursions of the COM (*Geyer, Seyfarth & Blickhan, 2006*). Inverted pendulum mechanics have been used to describe walking gaits, as the COM vaults over the grounded limb during the stance phase, which allows energy exchanges between $E_p$ and $E_k$ to occur (*Cavagna, Heglund & Taylor, 1977*). In running gaits, the COM descends whilst the limb is grounded due to limb compliance. This results in $E_p$ and $E_k$ being largely in phase, so instead energy is stored in long elastic tendons as the COM descends then returned later in the step (*Cavagna, Heglund & Taylor, 1977*; *Blickhan, 1989*). Spring mass or spring-loaded inverted pendulum (SLIP) models are often used to investigate running gaits (*Ruina, Bertram & Srinivasan, 2005*). Energy conservation by either inverted pendulum or spring mass mechanisms reduces the need for muscular work to replenish lost mechanical energy.

During the stance phase of the stride, the motion of the COM is redirected from forward-downward to forward-upward. Redirection of the COM contributes to the external work of locomotion and is one of the main sources of mechanical energy losses (*Bertram & Hasaneini, 2013*; *Bertram, 2013*). Collisional mechanics describes the action of the limbs in redirecting the COM through the generation of GRFs. The work done when a limb is in contact with the ground depends on the angular relationship between the COM velocity vector (**V**) and the resultant GRF vector (**F**) (*Lee et al., 2011*). The collision angle is calculated as the difference between **F** and **V** shifted by $\pi/2$. Up to 0.3 radians the collision angle is closely related to the mechanical cost of motion (*Lee et al., 2011*), which is a dimensionless metric of locomotor mechanical energy expenditure.

Considerable energy is expended when a horse's hooves push against the ground to redirect the path of the COM, so collisions are a significant source of energy losses during locomotion (*Bertram & Hasaneini, 2013*). Limb coordination strategies that minimize the

cost of reversing the motion of the COM from forward and downward at limb contact to forward and upward at lift off are favored. One of the ways in which this is achieved is through the sequencing of limb contacts with the ground, so that the COM velocity vector is gradually reoriented with each successive limb collision (*Ruina, Bertram & Srinivasan, 2005*). In this regard, the tölt with individual footfalls has energetic advantages over the diagonally-synchronized footfalls of the trot, as the work required to raise the COM in each stride is halved (*Usherwood, 2019*).

Each gait has a typical limb coordination pattern that may vary a little with speed and that changes abruptly during transitions between gaits. Typically, horses walk at slow speeds, trot at intermediate speeds and canter or gallop at fast speeds with each gait being used over a range of speeds that allows the cost of transport to be maintained at a low level (*Hoyt & Taylor, 1981*). Transitions between gaits, which involve changes in limb coordination, may be triggered by the need to reduce limb forces as speed increases (*Farley & Taylor, 1991*). In walking and galloping, each limb contacts the ground at a different time in the stride cycle which results in four collisions per stride. Having a larger number of footfalls smooths the COM trajectory and reduces collisional energy losses (*Ruina, Bertram & Srinivasan, 2005*; *Lee et al., 2011*; *Usherwood, 2019*). In diagonal gaits, such as trot, the synchronous diagonal contacts of fore and hind limbs result in higher collisional energy losses and more abrupt changes in COM trajectory compared with walk or gallop (*Lee et al., 2011*). Within a gait, limb sequencing can be adjusted to reduce collisional energy losses. Adjustments of this nature occur in trotting horses, where hind-first dissociation of the diagonal limb contacts has been associated with a significant reduction in mechanical energy losses compared with fully synchronous contacts (*Hobbs, Bertram & Clayton, 2016*).

In contrast to the parsimonious energetics of the natural gaits, the equestrian sport of dressage rewards the horse for performing with great activity and impulsion, without regard for energetic efficiency. The horse is trained to perform the natural gaits over an enhanced speed range and to change gaits in response to a cue from the rider thus over-riding the horse's natural inclination to make transitions based on physiological or biomechanical variables. Four types of trot are performed that differ not only in speed but also in the body posture and limb kinematics. In order of increasing speed (mean $\pm$ SD) they are collected trot ($3.20 \pm 0.28$ ms$^{-1}$), working trot ($3.61 \pm 0.10$ ms$^{-1}$), medium trot ($4.47 \pm 0.23$ ms$^{-1}$), and extended trot ($4.93 \pm 0.14$ ms$^{-1}$) (*Clayton, 1994*). In addition, highly trained dressage horses are taught to perform two, diagonally-coordinated artificial gaits. Passage is a very slow, majestic trot performed at speeds in the range of 1.2–1.9 ms$^{-1}$ (*Holmström, Fredricson & Drevemo, 1995*; *Weishaupt et al., 2009*; *Clayton & Hobbs, 2017*). It is distinguished by the fact that the limbs appear to hover at their most elevated position in the swing phase. Piaffe is performed with little or no forward movement with the limbs being raised and lowered in diagonal pairs (*Fédération Equestre Internationale, 2019*). Passage and piaffe represent the highest level of collection and self-carriage in competitive dressage.

This study investigates collisional mechanics in horses trained to maintain a diagonal limb coordination pattern over a wide range of speeds that extend beyond those performed naturally in order to investigate the spectrum of locomotor capabilities. The objective is to investigate relationships between mechanical cost, COM excursion and spring-mass limb

mechanics in diagonally-coordinated equine gaits across a range of speeds. The hypothesis is that, as speed decreases, the COM velocity will have a steeper vertical trajectory that will be associated with higher mechanical energy cost.

## METHODOLOGY

This study was performed with approval from the institutional animal care and use committee under protocol number 02/08-020-00 (Michigan State University, USA).

The subjects were three highly-trained dressage horses (mean ± SD, mass: 546 ± 9 kg) ridden by the same highly experienced rider (mass 61.5 kg including saddle).

### Data collection

Data were collected as the horses moved along a 40 m by 12 m rubberized runway. A series of four synchronized force plates were embedded in the middle part of the runway. The first and last plates measured 60 × 120 cm (FP60120; Bertec Corp., Columbus, OH, USA) and the middle two measured 60 × 90 cm (FP6090; Bertec Corp., Columbus, OH, USA). Force data were recorded at 960 Hz per channel. Successful trials were judged subjectively by a dressage trainer as those in which the horses moved straight and correctly for the gait being performed through the data collection volume.

Retroreflective markers attached to the skin were tracked using 10 infra-red cameras (Eagle cameras; Motion Analysis Corp., Santa Rosa, CA, USA) recording at 120 Hz and motion analysis software (Cortex 1.1.4.368; Motion Analysis Corp., Santa Rosa, CA, USA). A full body kinematic model as described in *Hobbs, Richards & Clayton (2014)* was constructed with the omission of trunk tracking markers T10 and T18, which provided a COM location for the trunk throughout the dynamic trials. Markers located at the centre of rotation of the distal interphalangeal joints were confirmed using radiographs.

Horses were familiarized with the research environment before commencing data collection and warmed up appropriately. Trials were performed at a range of speeds in the diagonally-synchronized gaits of trot, passage and piaffe in a random order. The horses trotted at a range of speeds to include trials at collected, working, medium and extended trot.

### Data processing and analysis
#### *All analyses were conducted using sagittal plane data*
The timings of hoof contacts and lift offs were identified from the force data using a threshold of 50 N, except for piaffe where centre of pressure (COP) excursion was used to detect overlapping limb contacts. Diagonal steps for the left forelimb and right hindlimb pair (LFRH) and the right forelimb and left hindlimb pair (RFLH) were extracted when they were available.

Kinematic data were smoothed with a 4th order Butterworth filter with 10 Hz cut off frequency and kinetic data with a 4th order Butterworth filter with 100 Hz cut off frequency. From these data, selected variables were calculated for each frame of each available diagonal step. Trunk COM location was extracted from the full body kinematic model and trunk velocity was then calculated from the first derivative. Limb length and inclination were
determined from the tuber spinae scapulae (forelimb) and greater trochanter (hindlimb) to the centre of rotation of the distal interphalangeal joint. Trunk COM vertical excursion was calculated as the difference between minimum and maximum heights for each step. Limb compression, calculated as the difference in vertical limb length during standing and minimum vertical limb length during the stance phase, was normalized to a percentage change from the standing length. The range of limb pro-retraction was calculated as the difference between the maximum protraction to maximum retraction angles of the limb.

Vertical and longitudinal GRFs were summed over the fore- and hind limbs, and the resultant force vector $\mathbf{F}$ was calculated from the summed components. Similarly, the resultant trunk velocity vector $\mathbf{V}$ was calculated from its vertical and longitudinal components. The instantaneous collision angle ($\varphi$) was calculated from the dot product of force on velocity and the collision angle ($\Phi$) was determined by force and velocity averaging over each step (*Lee et al., 2011*). The mechanical cost of motion ($CoMot_{mech}$) was calculated for each step as the sum of the dot product of force and velocity divided by the sum of absolute force multiplied by absolute velocity (Eq. (1) from *Lee et al., 2011*).

$$CoMot_{mech} = \Sigma|\mathbf{F}\cdot\mathbf{V}|/\Sigma|\mathbf{F}||\mathbf{V}| \tag{1}$$

To calculate summary vector variables (*Hobbs, Robinson & Clayton, 2018*), GRF data were normalized to horse mass. VecMag was calculated by vector summation of the individual vectors divided by the number of samples contributing to the value. VecAng was determined trigonometrically from the components of the vector magnitude and expressed relative to the vertical with positive values being directed cranially. GRFs were down-sampled to 120 Hz and GRF vector diagrams were constructed as examples of each gait.

As collision angles >0.3 rad were expected at slower speeds, linear regression was used to investigate the relationship of $CoMot_{mech}$ and speed across gait classifications. Partial correlation controlling for horse was then used to examine the relationships between COM excursion and limb mechanics for each gait classification separately. Significance was set at $p < .05$.

## RESULTS

The range of gaits were classified by speed with no overlap between groups; piaffe <0.6 ms$^{-1}$, passage 0.8 to 1.8 ms$^{-1}$ and trot 2.0 to 5.12 ms$^{-1}$. Metrics obtained for calculated variables separated by gait classification and with trot data separated by speed are presented in Table 1.

### COM

For all gaits at all speeds the trunk COM followed the trajectory of a spring-mass model associated with bounding gaits; the highest points of the diagonal stance phase were at contact and lift off and the COM descended during the middle of stance (Fig. 1). For piaffe, passage, slow trot and fast trot, respectively, maximum height of the COM during each gait compared to standing height was $18 \pm 15$, $42 \pm 21$, $5 \pm 17$, $-26 \pm 20$ mm and minimum

**Table 1 Measured variables for piaffe, passage, slow trot and fast trot.**

| | Piaffe | Passage | Slow trot $<3.5$ ms$^{-1}$ | Fast trot $\geq 3.5$ ms$^{-1}$ |
|---|---|---|---|---|
| $n$ | 65 | 26 | 73 | 24 |
| Speed (ms$^{-1}$) | 0.19 (0.16) | 1.27 (0.19) | 2.68 (0.39) | 4.37 (0.57) |
| Collision angle $\Phi$ (rad) | 0.88 (0.33) | 0.38 (0.04) | 0.27 (0.02) | 0.26 (0.01) |
| CoMot$_{mech}$ | 0.73 (0.21) | 0.34 (0.03) | 0.21 (0.02) | 0.19 (0.01) |
| COM$_v$ excursion (mm) | 68.7 (20.8) | 111.1 (17.3) | 77.0 (15.0) | 86.4 (18.1) |
| $\Delta$length$_F$ (%) | 1.47 (2.06) | 2.94 (2.54) | 4.46 (0.91) | 6.99 (1.77) |
| $\Delta$length$_H$ (%) | 7.81 (1.93) | 6.39 (1.06) | 5.17 (1.33) | 8.53 (2.13) |
| Pro-retraction$_F$ (deg) | 5.1 (3.1) | 22.3 (4.7) | 33.1 (5.0) | 39.5 (4.7) |
| Pro-retraction$_H$ (deg) | 5.9 (2.7) | 24.7 (2.8) | 36.2 (2.8) | 44.6 (2.7) |
| VecMag$_F$ (N/kg) | 41.2 (2.5) | 47.7 (3.5) | 51.4 (4.2) | 63.9 (7.0) |
| VecMag$_H$ (N/kg) | 35.0 (2.6) | 38.4 (4.8) | 38.1 (2.6) | 46.9 (5.0) |
| VecAng$_F$ (deg) | 0.12 (1.61) | $-4.02$ (1.33) | $-1.59$ (1.35) | $-1.16$ (1.15) |
| VecAng$_H$ (deg) | 0.37 (2.20) | 3.72 (2.86) | 2.84 (1.62) | 6.06 (2.41) |

**Notes.**
Mean (s.d.) for collision angle ($\Phi$) mechanical cost of motion (CoMot$_{mech}$), centre of mass vertical excursion (COM$_v$ excursion) limb mechanics ($\Delta$length$_{F,H}$; Pro-retraction$_{F,H}$) and summary vector variables (Mean vector magnitude in the forelimbs and hind limbs: VecMag$_{F,H}$; Mean vector angle in the forelimbs and hind limbs: VecAng$_{F,H}$) separated by gait classification. To illustrate the influence of speed in trotting, trot has also been separated into above and below 3.5 ms$^{-1}$.

height compared to standing height was $-51 \pm 22$, $-69 \pm 24$, $-71 \pm 15$, $-113 \pm 18$ mm. During trotting, the path of the COM became longer and lower as speed increased.

## Sagittal plane GRF vectors

Vector diagrams representing the sagittal plane GRFs for the left and right diagonals in fast and slow trot, passage, and piaffe (Fig. 1D) show markedly lower vertical force magnitude, represented by the height of the force vectors, in both fore and hind limbs for piaffe. The horizontal spread of the force vectors represents the longitudinal forces. A wide spread of force from braking to propulsion is evident in trot. Compared with trot, passage has higher magnitudes of hind limb propulsive force and forelimb braking force and piaffe has a much smaller spread of longitudinal force in both limbs. Force-time graphs representing mean vertical and longitudinal forces for the RFLH diagonal step in each gait are shown in (Fig. 1C). VecMag values and vertical GRFs are considerably lower in piaffe compared with the other gaits (Table 1, Fig. 1).

## Collision angle

The instantaneous collision angle represents the deviation from a perpendicular relationship between the COM velocity vector and GRF vector throughout stance (Fig. 1B). In trot the instantaneous collision angle falls and rises once during each diagonal step. The value falls to approximately zero in the middle part of stance in trotting coinciding with the time when the GRF vector and velocity vector are perpendicular. The first and second halves of the curve are almost mirror images of each other. The shape of the instantaneous collision angle curves are similar in passage and piaffe but the values in early and late stance get progressively higher. In piaffe steps the fore and hindlimb contacts and lift offs overlap. During these periods of overlap the vertical GRFs are small and there is a rapid decrease

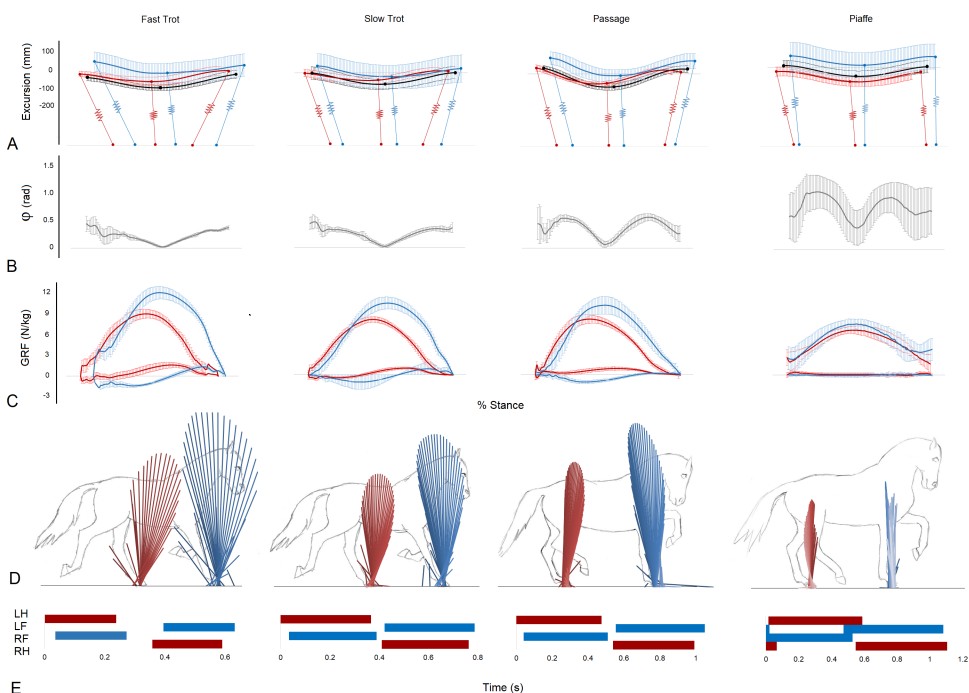

**Figure 1** Data illustrating the mechanics of each gait taken from the RFLH diagonal (mean ± standard deviation) (A–C), one RFLH step (D) and continuous RFLH-LFRH strides (E). (A) Vertical excursion compared to standing (mm) of the trunk COM (black), forelimb represented by scapula COM (blue) and hindlimb represented by the greater trochanter of the femur (red). Approximate limb inclinations are illustrated at footstrike, midstance and lift off. Forelimbs and hindlimbs are offset on the horizontal axis to position them cranially and caudally, respectively, from the COM. (B) Instantaneous collision angle (rad). (C) Forelimb (blue) and hind limb (red) ground reaction forces (N/kg). (D) Vector diagrams for the fore-limbs (blue) and hindlimbs (red). (E) Footfall sequences and timing (s).

in the instantaneous collision angle as the velocity and GRF vectors are closer to being orthogonal (Fig. 1B). The collision angle Φ for 48% of the steps was >0.3 rad. These steps were mainly found at piaffe and passage (Table 1).

## Mechanical cost

The relationship between speed and mechanical cost for each gait is illustrated in Fig. 2, where significant relationships were found for piaffe ($R^2 = 0.799$, $p < .001$) and trot ($R^2 = 0.333$, $p < .001$). In piaffe, a small forward speed resulted in relatively low mechanical cost of motion but this increased steeply as speed slowed to zero. Within passage, speed did not influence mechanical cost significantly. During trotting, a slight but significant decrease in mechanical cost was found with increasing speed.

## Spring-mass mechanics

Relationships between trunk COM vertical excursion and limb mechanics varied with gait classification (Table 2). For all gaits, significant relationships ($p < .05$) were found between COM vertical excursion variables and both fore and hind limb compression, with stronger and consistently more significant ($p < .01$) relationships to the forelimb.
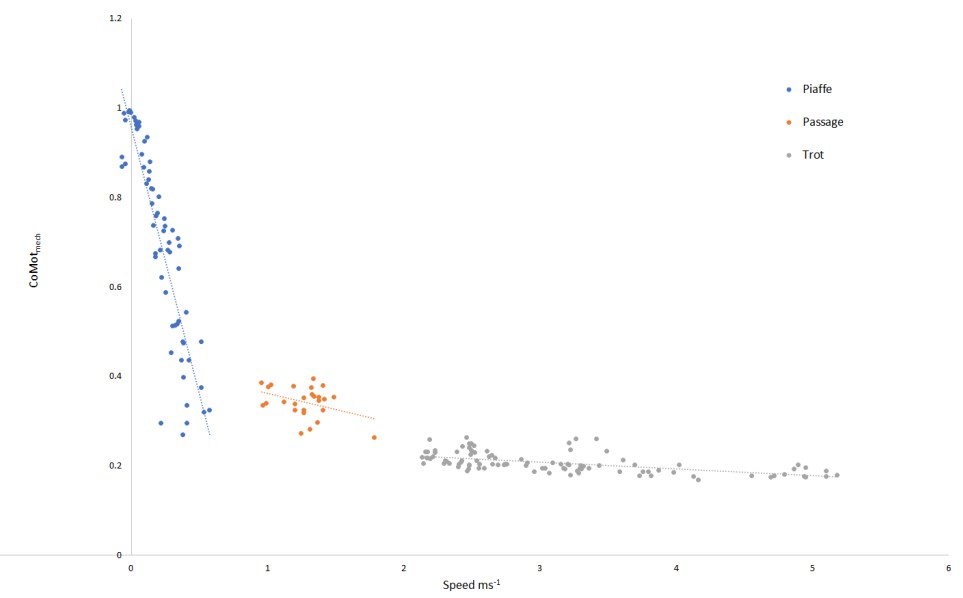

**Figure 2** **Relationship between speed and mechanical cost of motion for each gait classification.** Key: piaffe, blue; passage, orange; trot, grey.

**Table 2** **Results of Partial correlation (controlling for horse) between COM vertical excursion and minimum height and limb mechanics.**

| | | $\Delta length_F$ | $\Delta length_H$ | Pro-retraction$_F$ | Pro-retraction$_H$ |
|---|---|---|---|---|---|
| Piaffe | $COM_v$ excursion | 0.628** | 0.582** | 0.002 | −0.143 |
| | $COM_v$ min | −0.725** | −0.667** | 0.057 | 0.160 |
| Passage | $COM_v$ excursion | 0.731** | 0.467* | 0.223 | 0.007 |
| | $COM_v$ min | −0.523** | −0.486* | −0.033 | 0.016 |
| Trot | $COM_v$ excursion | 0.486** | 0.303** | −0.057 | 0.273** |
| | $COM_v$ min | −0.877** | −0.721** | −0.451** | −0.806** |

Significant correlations **$p < .01$, *$p < .05$.

Limb pro-retraction only influenced COM excursion and height at trot, with greater pro-retraction of the forelimb or hind limb the minimum height of the COM was lower. This was more pronounced in the hindlimb ($p < .01$) and as hindlimb pro-retraction increased, COM vertical excursion also increased ($p < .01$).

## DISCUSSION

In this study the relationships between mechanical cost of motion, COM excursion and spring-mass limb mechanics in diagonally-coordinated equine gaits were explored across a range of speeds. The mechanical cost of motion was found to increase significantly with a decrease in speed in trotting and piaffe, with the effect being most pronounced when forward speed was close to zero. Overall, as speed decreased, the horizontal component of COM velocity reduced, and as a consequence the velocity vector **V** became closer to the vertical. At the same time, limb movements and the force vector **F** were directed more

vertically. Convergence of the vectors **V** and **F** resulted in an increase in the mechanical cost of each collision with the ground. Therefore, the hypothesis is supported.

Collisional mechanics provide an elegant method of comparing the cost of locomotion between gaits and across species, since the values are dimensionless. As a diagonally coordinated gait, the mechanical cost of trotting in dogs and goats is higher than walking and galloping, due to the close sequencing of landing and lift-off of the fore and hindlimbs in each trotting step. Values are reported to be 0.031 and 0.074 for walking, 0.212 and 0.192 for trotting, and 0.059 and 0.105 (rad) for galloping, for dogs and goats respectively (*Lee & Biewener, 2011*; *Lee et al., 2011*). In this study, the collision angles at trot are higher than those reported for dogs and goats and increase further for passage and piaffe. In addition to differences between species, the presence of a rider may affect the movement patterns and energetics both through the gravitational and inertial effects of the rider's weight and through the rider's effect on the horse's posture and performance. Adding weight to the horse's back hollows the thoracolumbar spine (*De Cocq, Van Weeren & Back, 2004*; *De Cocq et al., 2009*) and may change the weight distribution between the fore and hind limbs (*Schamhardt, Merkens & Van Osch, 1991*; *Clayton et al., 1999*; *Licka, Kapaun & Peham, 2004*), potentially altering limb loading and COM excursion. It is not known how the presence compared to the absence of a rider affect mechanical energy exchanges.

The high mechanical cost in trotting can be alleviated somewhat by dissociation between fore and hindlimb landings, with hind first landings reducing energy losses more than fore first landings in horses (*Hobbs, Bertram & Clayton, 2016*). The duration of hind-first dissociation increases with trotting speed (*Hobbs, Bertram & Clayton, 2016*), and a relatively long hind-first dissociation is also a feature of passage (*Holmström, Fredricson & Drevemo, 1995*; *Clayton, 1997*; *Clayton & Hobbs, 2017*). In passage, any mechanical advantages gained by dissociation of the diagonal limb contacts were likely overridden by the effects of the slower speed and increased COM vertical excursion compared to trot. Greater energy losses are found in larger down to up deflections of the COM (*Bertram & Hasaneini, 2013*). During forward-moving trials of piaffe, although the speed was slower than passage the energy costs in the two gaits were similar. Fore first landings are more commonly found in piaffe (*Holmström, Fredricson & Drevemo, 1995*; *Clayton, 1997*), but also in this study many of the steps had overlapping contacts and lift offs. The similar costs for passage and forward-moving piaffe steps may be a result of overlapping steps that potentially suggest a greater use of limb sequencing to minimize cost and/or smaller down to up COM vertical excursions (*Ruina, Bertram & Srinivasan, 2005*; *Usherwood, 2019*). As speed tends towards zero in piaffe, energy is only required for upward motion, the velocity and GRF vectors become more closely aligned, and the cost of motion rises steeply. The rimless rolling wheel representing the collisional effects of successive footfalls (*Coleman, Chatterjee & Ruina, 1997*) no longer rolls and the limbs have a vertical trajectory. Piaffe on the spot is possibly more akin to modelling a bouncing ball or a human hopping on the spot (*Blickhan, 1989*), which may require different energy conservation strategies.

During trotting $E_k$ and $E_p$ are largely in phase and so energy is conserved through elastic strain energy storage in the spring elements of the limbs and trunk (*Cavagna, Heglund & Taylor, 1977*). Any energy lost to the system, such as the losses during each

collision must be replaced by muscle work (*Biewener, 2006*). In piaffe, energy exchange is expected to be largely between vertical $E_k$, $E_p$ and strain energy. Piaffe steps are slower than trotting steps (*Holmström, Fredricson & Drevemo, 1995*; *Clayton, 1997*), the shoulder and hock joints are continually more flexed during the stance phase than in other diagonal gaits, and the fore and hind fetlock joints extend less (*Holmström, Fredricson & Drevemo, 1995*). The reduction in fetlock extension is likely due to the lower vertical forces on the limbs (*McGuigan & Wilson, 2003*), which is illustrated by lower VecMag values here and probably explains why there is little change in overall forelimb length during piaffe steps compared to standing length. The range of motion of segments and joints during stance is also generally smaller in piaffe, which led *Holmström, Fredricson & Drevemo (1995)* to conclude that elastic strain energy may not be important. Similar to piaffe, humans hopping on the spot maintain flexion of the limb joints, which increases limb compliance and allows for greater energy storage if the hopping frequency is low (*Blickhan, 1989*). In addition, most of the limb muscle work is isometric so changes in limb length are mainly taken up by lengthening of the tendons (*Blickhan, 1989*). Our results illustrate that COM vertical excursion is strongly affected by limb length changes, which suggests that tendon lengthening and strain energy storage is a feature of piaffe, similar to hopping. This may be more important in the hindlimb, as limb compliance is greater due to the greater change in length compared to standing. If a large amount of the available mechanical energy is lost in collisions with the ground when performing a stationary piaffe due to aligned GRF and velocity vectors, reducing the step frequency and increasing hindlimb compliance to enhance energy storage in the tendons may be an important energy conservation strategy. In addition, isometric and/or concentric muscle force production may be necessary to elevate the COM in each successive step.

In contrast to the near-vertical limb orientation in piaffe, pro-retraction pendular motion was important to COM vertical excursion in trot, together with spring compression. The range of inverted pendulum motion increased with speed, indicating a longer step length and the VecMag values are indicative of greater force production and consequently greater limb spring compression (*Biewener, 2006*). These data conform well to the SLIP model (*Ruina, Bertram & Srinivasan, 2005*; *Lee et al., 2011*). In our study, the combination of increased limb compression and greater pendular motion in trotting reduced the overall height of the COM as speed increased, but with increased vertical COM excursion. A reduction in COM excursion may be more efficient in human running, but not at the expense of greater limb work to smooth the up-down motion (*Bertram & Hasaneini, 2013*). The gradual reduction in cost of motion reported here as trotting speed increased is expected to be the result of increased diagonal dissociation (*Hobbs, Bertram & Clayton, 2016*) and increased hindlimb pendulation with increasing speed, which orients the summed GRF vector, F closer to the vertical during the absorbing phase. Additional limb work may be required to achieve this (*Bertram & Hasaneini, 2013*), so the net energy in fast trot may be greater, but this is currently not known. In contrast, the role of the forelimb in passage is to support the COM in a higher position at the start of stance which, in effect, increases vertical COM excursion, as reported previously by *Clayton & Hobbs (2017)*. Since the forelimb acts principally as a passive limb spring

(*McGuigan & Wilson, 2003*), the elevation appears to be achieved by limiting limb pro-retraction, which then influences weight distribution between the fore and hind limbs (*Clayton & Hobbs, 2017*).

This study compared data from three Lusitano Grand Prix horses performing the three gaits of trot, passage and piaffe. This limited our methods of statistical analysis and generalization to all dressage horses performing these gaits should be made with caution.

## CONCLUSIONS

This study has shown a negative relationship between velocity and collisional energy losses in trot and piaffe. In general, a decrease in horizontal velocity of the trunk COM caused the velocity vector to become more closely aligned with the GRF vector resulting in an increase in the mechanical cost of each collision with the ground. This was particularly evident in piaffe, in which, as speed approached zero, the COM vector and the GRF vector were both oriented almost vertically throughout stance and collisional energy losses increased more than threefold compared with steps that were moving forward at 0.6 m/s. Vertical excursions of the trunk COM were strongly correlated to changes in limb spring lengths, which were more pronounced in the forelimbs in all gaits. In piaffe, increasing hindlimb compliance is probably an important energy conservation strategy.

### Funding
The authors received no funding for this work.

### Competing Interests
Hilary M. Clayton is the CEO of Sport Horse Science.

### Author Contributions
- Sarah Jane Hobbs conceived and designed the experiments, analyzed the data, contributed reagents/materials/analysis tools, prepared figures and/or tables, authored or reviewed drafts of the paper, approved the final draft.
- Hilary M. Clayton conceived and designed the experiments, performed the experiments, contributed reagents/materials/analysis tools, authored or reviewed drafts of the paper, approved the final draft.

### Animal Ethics
The following information was supplied relating to ethical approvals (i.e., approving body and any reference numbers):

This study was performed with approval from the Institutional Animal Care and Use Committee (Michigan State University, USA) under protocol number 02/08-020-00.

### Data Availability
The raw data are available in the Supplemental File.
**Supplemental Information**

Supplemental information for this article can be found online at http://dx.doi.org/10.7717/peerj.7689#supplemental-information.

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
