# Peer review of "Collisional mechanics of the diagonal gaits of horses over a range of speeds"

_PeerJ, doi:10.7717/peerj.7689_

## Round 0.1 · original submission · Minor Revisions

Apologies for some delay in obtaining reviews- it seems summertime has caught us with too few reviewers available. But we now have 3 reviews for this MS and all raise some points of use in revising the MS. Overall, the changes are moderate but I agree that some more engagement with the rather extensive literature on collisional mechanics in locomotion would improve the paper, as would some changes to figures as recommended by multiple reviewers. Please address all points individually in your Response. Thank you!

·

Basic reporting

No comment

Experimental design

No comments

Validity of the findings

While underlying data have been provided - documentation of the definitions of data in a few columns are unclear - e.g., Import QQ sheet, B/F?
Where applicable - include units in underlying data

Additional comments

Figure 1 – consider putting parts a, b, and c on the same horizontal time axis – so that the differences in stance time are evident

Figure 1 – consider adding footfall sequence for each movement – e.g., piaffe – the fore and hind would overlap some

Line 239 – affected

Reviewer 2 ·

Basic reporting

This report describes a small but well conducted study of the specifics of three somewhat related equine dressage competition gaits that are somewhat related; the passage, piaffe and trot. In particular the objective was to assess these gaits in terms of collision dynamics, an emerging perspective on gait movement strategy and its energetic consequences. Although only a small number of animals were assessed, these were highly trained animals well schooled in these gaits. The relatively small variability of the data indicate that the results are indicative of the general behaviour of these animals and the results can be depended upon. Table 1 and Fig. 2 alone justify publication of this paper, but the question addressed is provided within a solid background of theory and relevant literature. This paper will be of substantial interest to a relatively small group of individuals who focus on equine gaits, particularly those of dressage. This paper provides a novel perspective and introduces a new, and in my opinion valuable, method of comparing the dynamics of these gaits.

Experimental design

Line 89 re: data recording rate - is this 960 Hz in total or per channel? This should be specified.

Line 90 What is meant by the term ‘consistently’? Did you measure horizontal acceleration? Was speed consistent stride to stride?

Validity of the findings

The findings are valid and the authors do a good job in describing the value of the findings and their implications.

Additional comments

Lines 245-248 I don’t understand this statement. At the trot the system should act like a spring-mass combo, not an inverted pendulum. If you are invoking the more complex spring-loaded inverted pendulum (SLIP) model, the statement would make some sense, but this should be clearly described and specified.

Reviewer 3 ·

Basic reporting

Overall, I do not feel the literature is entirely well referenced. There is opportunity here to compare these data to other gaits and other species. This should be considered: Usherwood, J. R. (2019). An extension to the collisional model of the energetic cost of support qualitatively explains trotting and the trot–canter transition. Journal of Experimental Zoology Part A: Ecological and Integrative Physiology.

Experimental design

No Comment

Validity of the findings

No comment

Additional comments

This paper makes use of a series of force plates and a motion capture system to calculate the mechanical cost and associated collision variables in ridden equine trot, passage and piaffe. It would appear these data have been collected to a high technical and ethical standard and the methods are described sufficiently to allow replication. The underlying data have been provided as values extracted from the raw data files which is sufficient given the nature of this type of data.

I believe this manuscript is a valuable and worthwhile contribution to the literature though work is needed to make this clear to the reader and to maximise its usefulness. I actually think the authors sell themselves short and the manuscript would really benefit from a more thorough introduction and discussion. In particular, I feel that there needs to be more context and more reference to the literature. I also feel strongly that the figures are somewhat thin on the ground and, again, much more could be made of this dataset (I am unsure whether the authors were limited by the journal in this or whether they have plans for further manuscripts using the same data).

This work is interesting and I would be very happy to see it published but I think the authors need to revise their introduction and discussion to make this clear to all readers.

Abstract:
I understand there is a word limit here but I think there needs to be a sentence at the end to hit home the relevance, given this is what readers will use to decide whether they read the rest of the paper.

Introduction:
The introduction currently reads like an excerpt from a biomechanics textbook, it gives a very specific introduction to collisions but lacks broader context. I appreciate the authors have a tricky job here but this paper is actually interesting and useful to both pure biomechanists and equine specialists - this is very valuable impact-wise but a juggling act to please all parties. Specifically, I think the introduction would very much benefit from an early paragraph discussing broadly the cost of locomotion and its importance followed by how manipulating gait (ie. gaits used in horse riding that don’t exist in nature) can provide valuable insight into underlying mechanics and mechanical cost. As this is not a specialist equine journal, I do think it is necessary to fully describe and give background to the passage and piaffe gaits (similarly, working, medium and extended trot would need to be explained to a non-equine audience). I hope the authors do not take this as too heavy a criticism as it is not intended to be, I simply think this interesting dataset could be better framed and be more impactful as a result.

Methods:
The methods are clearly described and repeatable. I have a few questions that I did not pick up answers to when reading and apologise if I missed them:
- horse and rider weight are given to 1 decimal place but I did not see mention of the weight of the tack, was this considered?
- were the data checked for acceleration/deceleration within trials. I fully appreciate that perfect steady state is impossible to achieve but this would make a difference to the data, a sentence mentioning how this was controlled for/checked would be useful.
- the range of speeds in the piaffe especially is very small, do you have any metric for how accurately this was measured eg. over what distance/time?

Results:
As above, I think more could be made of the data with the use of figures.
- In figure 1C, it would be nice to see separate forelimb and hindlimb traces either in addition to the summed or instead of. Also, it would make sense to show a mean in all panels (+/- std) for a mid-range speed or at least one from either end of the speed range to give an idea of variability rather than a single step.
- As you discuss energy exchange a bit later on, would it be possible to show PE, KE etc. throughout a stride for each gait? This would be of great interest to those studying cost of locomotion and would add significant weight to the discussion points. These points come across as largely speculative but you have the data available to support these statements.
Overall, I can fully appreciate that these will have been challenging data to collect so showing it off with greater use of figures would make sense.

Discussion:
As with the introduction, I believe this can be extended to be more impactful.
- 208: you mention the effect of the rider, is it possible to suggest how the results might be affected by the rider? I appreciate these data could not be collected without a rider but some discussion/quantification would be beneficial for the reader.
- 211: at what point of dissociation is it no longer a trot? Is there any study into the limb phasing here?
- Reference to other gaits is limited, there is a wealth of literature on equine gait, I feel comparison with other gaits would really help contextualise these data for the reader. For example, how does the cost of passage compare to a canter or tolt?
- Overall, I do not feel the literature is entirely well referenced. There is opportunity here to compare these data to other gaits and other species. This should be considered: Usherwood, J. R. (2019). An extension to the collisional model of the energetic cost of support qualitatively explains trotting and the trot–canter transition. Journal of Experimental Zoology Part A: Ecological and Integrative Physiology.

Minor points:
Line 16: maybe “loss” would read better than “losses”?
Line 37: “from place to place”… “in relation to the environment”?
Line 39: need a more basic reference here.
Line 40: not really a goal, more worthwhile to say there is benefit to minimising energy expenditure for given movement.
Line 41: ‘Method of achieving’ sounds a bit like there’s some conscious choice, also worth mentioning that these are highly reductionist etc.
Line 53: need some references here as a number of equine gaits are shown to be relatively inexpensive.
line 55: true in gallop, maybe walk but is this shown in trotting gaits given they are the focus of this paper? Otherwise need to specify.
Lin 62: wording here is a little ambiguous, a speed exists where cost is minimised, moving further away from this speed increases cost.
Line 53/64: one of the theories.
Line 111: comma after data.
Line 129: maybe an editing point but the bars in the equation have come out as capital i which is confusing.

---

## Round 0.2 · accepted · Accept

I have checked the Rebuttal and MS and feel that the reviews have been nicely taken on board, with substantial improvements to the MS, so I see no need for further review/revision. Congratulations!

#